# Susceptibility of domain experts to color manipulation indicate a need for design principles in data visualization

**Markus Christen** [1,2]*, **Peter Brugger**[3,4], **Sara Irina Fabrikant**[2,5]

**1** Institute of Biomedical Ethics and History of Medicine, University of Zurich, Zurich, Switzerland, **2** Digital Society Initiative, University of Zurich, Zurich, Switzerland, **3** Department of Psychiatry, University Hospital Zurich, Zurich, Switzerland, **4** Rehabilitation Center Valens, Valens, Switzerland, **5** Department of Geography, Geographic Information Visualization & Analysis Unit, University of Zurich, Zurich, Switzerland

* christen@ethik.uzh.ch

**Data Availability Statement:** All data files are available from the ZENODO database (DOI: 10. 5281/zenodo.4284817).

**Funding:** Cogito Foundation (Wollerau, Switzerland) under grant no. R-129/09 (M.

## Abstract

Color is key for the visual encoding of data, yet its use reportedly affects decision making in important ways. We examined the impact of various popular color schemes on experts' and lay peoples' map-based decisions in two, geography and neuroscience, scenarios, in an online visualization experiment. We found that changes in color mappings influence domain experts, especially neuroimaging experts, more in their decision-making than novices. Geographic visualization experts exhibited more trust in the unfavorable rainbow color scale than would have been predicted by their suitability ratings and their training, which renders them sensitive to scale appropriateness. Our empirical results make a strong call for increasing scientists' awareness for and training in perceptually salient and cognitively informed design principles in data visualization.

## Introduction

Color plays a pivotal role for the communication of scientific insights across many disciplines, particularly those that strongly rely on data-driven visualization, including geography and neuroscience [1]. Design principles and appropriate color schemes to depict data on maps—advanced by cartographers over centuries [2, 3]—are essential for effective visualization. Even though there are differences in visualization practices, standards, and design guidelines across disciplines [4, 5], some—such as neuroimaging [6, 7]—lack standards. Moreover, problematic scales such as the rainbow color scheme are still common across the sciences [8–11].

Despite such lacks in standards, color is widely applied to visually communicate research results [1, 3, 12, 13]. Color can alert viewers about outliers, highlight missing data, or signal unexpected relationships in a visual display. It also helps to group, to categorize or to order similar data values in multivariate data displays [10, 12–18]. However, improper color use might distract or, worse, mislead viewers, due to perceptual limitations of certain user groups (e.g., people with color deficiencies), color contrast effects, or neglect of cultural conventions. Finally, unintended emotional connotations, perceptual, and cognitive overload resulting from multi-colored data displays may rather confuse than assist the viewer and decision

Christen) - European Research Council, GeoViSense Project under grant number 740426 (S. Fabrikant).

**Competing interests:** The authors have declared that no competing interests exist.

maker; the rainbow color scheme applied widely in science is considered particularly problematic [11, 19–24].

Given the importance of color for data exploration, information visualization, and the communication of research findings in science, we set out to systematically assess its influence on trusting and interpreting visual information presented in data displays. We empirically studied this for four widely used color scales, and for two variations of typical display background colors (Fig 1A). We were also interested in how the visual interpretation of data might be affected by its domain and the associated expertise of the decision maker. For that, we employed two data contexts that could be solved across domain expertise levels, and including domain novices: brain activity states measured with positron emission tomography (PET), and ecosystem states in a country, derived from climate change model predictions (Fig 1B). We recruited neuroimaging experts, geographic information visualization experts (geovis), and lay people to participate in a web-based study with a between group design. Each participant solved one out of eight randomly assigned visualization conditions per data context (i.e., brain state or ecosystem state).

The experiment involved two map-based tasks, 1) assessing *trust* in the information conveyed by a visualization, namely whether the participants believe that data presented in a certain type of visualization supports an empirical claim, and 2) *data interpretation with a visualization*, namely, the ranking of a state, given two extreme states (Fig 1B; see methods for details). We define the multi-faceted concept of 'trust' for this study as the validity of a user's conclusions made from a data display. Additional questions concerned attitudes towards the phenomenon displayed by the visualization (i.e., brain state and ecosystem state) and respondents' professional experience and visualization production practices (lay people only provided basic information). We performed group comparisons after pair-matching participants regarding age, gender and data visualization expertise, resulting in 134 participants per group.

We hypothesized that; (a) domain experts have the lowest variability both in the trust and the data interpretation task; (b) response variability would decrease with increasing experience in visualization practices; (c) domain experts' opinions within their own data context (e.g., whether brain death indeed represents the death of a person) would not influence response variability, and that (d) geographic information visualization experts would show greater awareness on sound design principles compared to neuroimaging experts. Details of the methodology are presented in the section Materials and Methods (see below).

Contrary to our expectation, we found that changes in color mappings influence domain experts, especially neuroimaging experts, more in their decision-making than novices. Geographic visualization experts exhibited more trust in the unfavorable rainbow color scale than would have been predicted by their suitability ratings and their training. Our results make a strong call for increasing scientists' awareness for and training in perceptually salient and cognitively informed design principles in data visualization.

## Materials and methods

### Study population and pair matching

The study involved three groups: professionals in neuroimaging, professionals in geographic information visualization, and lay people. For identifying professionals in neuroscience, we determined all available e-mail-addresses of authors in papers published in *NeuroImage*, *Human Brain Mapping*, *Brain Imaging and Behavior*, *American Journal of Neuroradiology* (excluding papers on computer tomography) and *Annals of Neurology* (only neuroimaging papers) between 2012 and 2014. After deleting duplicates and excluding invalid e-mail-addresses, a set of 1491 e-mail-addresses remained. In total, 134 neuroimaging experts

# a) Experimental conditions

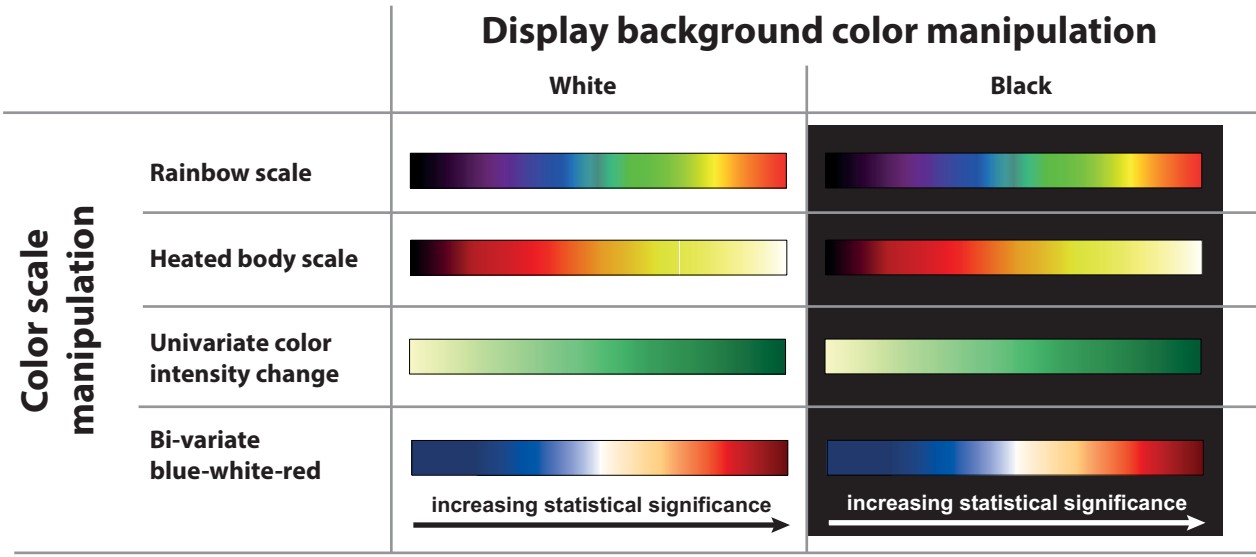

# b) Experimental tasks

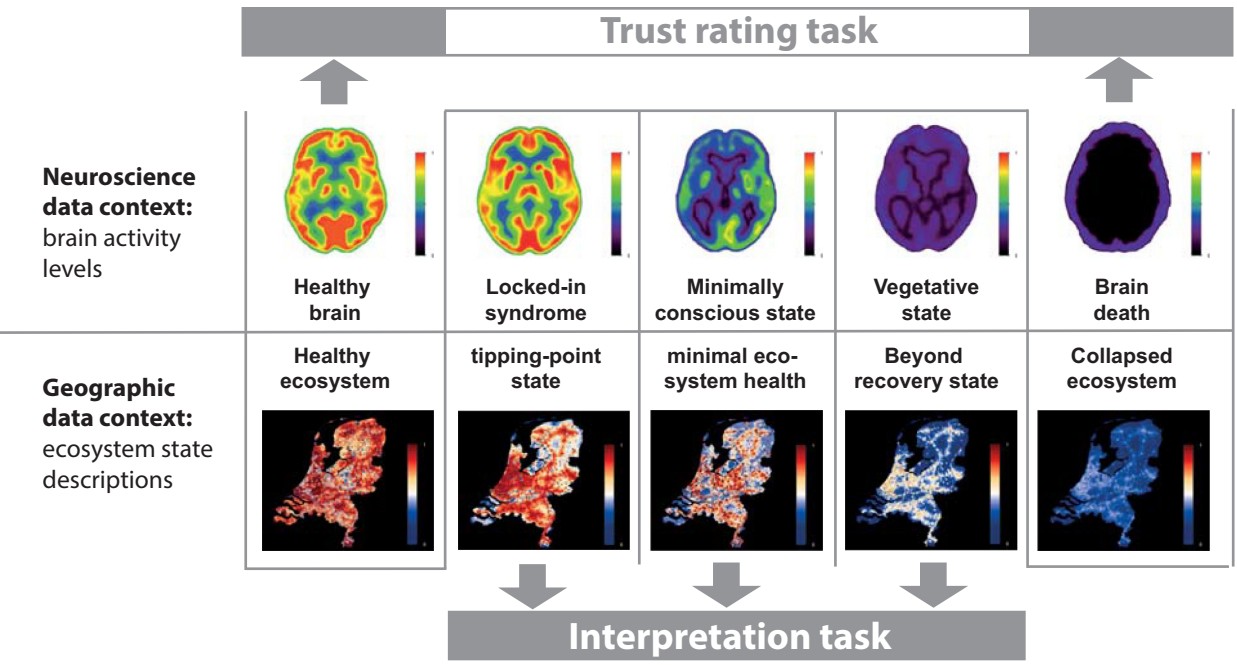

**Fig 1. Outline of the experiment.** We used eight experimental conditions a): 4 color scales (rainbow, heated-body, univariate color intensity, bi-variate red-white-blue) × 2 image background conditions (black and white). Experimental conditions remained unchanged for a single data context. b) Participants were presented with a neuroscience (i.e., brain activity states) and a geographic (ecosystem states) data context: First, participants had to perform a *trust rating task* for two extreme states ("healthy brain|ecosystem" and "dead brain|ecosystem"); then they had to perform an *interpretation task* by ranking three randomly presented intermediate states between the two extreme states (one condition for each data context).

completed the whole survey, yielding a response rate of 9.0%. For identifying professionals in geographic information visualization (geovis), we used the same procedure as for the neuroimaging experts by searching in the following journals: The *Cartographic Journal*, *Cartographic Perspectives*, *Cartographica*, *Cartography and Geographic Information Science*, *Computers & Geosciences*, *IEEE Transactions on Geoscience and Remote Sensing*, *International Journal of Geographic Information Science*, *ISPRS Journal of Photogrammetry and Remote Sensing*, *Journal of Maps* and the *Kartographische Nachrichten*. This search yielded 574 e-mail-addresses. In addition, electronic mailing lists of the International Cartographic Association (ICA) were also targeted, with several hundred registered members. As we cannot determine the overlap between the e-mails identified in the journals and the mailing list e-mails (the latter were not made accessible to us), we only can make a rough estimate of the total sample of approached geovisualization experts, if it is of the same order as the neuroscience experts (i.e., around 1500). In total, 197 participants completed the whole survey, yielding an estimated response rate of 13%. We carefully checked all responses for potential duplicates (e.g., by checking IP addresses and response patterns). For approaching lay people, we used Amazon Mechanical Turk (origin of the participants: USA). Participants were paid $1.20, average response time was 10 minutes (experts did not receive any payment). Here, the complete data of 486 participants were obtained. Nine (neuro), respectively ten (geo) persons of this group claimed to be neuroimaging or geovisualization experts–those filled out the according additional expert questions.

For pair matching with respect to age, gender (binary variable) and experience, the neuroimaging experts served as baseline. For assessing experience, the following variables were used: years of experience in neuroimaging (geography/geoscience, respectively), whether the participants consider themselves to be more a consumer or producer of images (binary variable) and whether the lab in which the participant is working is producing images or not (binary variable). For pair matching, we first created sub-groups of all combinations of the binary variables. Then, for the geovis experts, we randomly selected participants from that group, such that similar distributions with respect to age (neuro: 40.9, geo: 39.5; difference n.s.) and years of experience (neuro: 12.7, geo: 13.5, difference n.s.) were obtained. The lay people were pair-matched with respect to age and gender.

## Image creation

In total, five images per research domain (neuroscience and geography) were color-manipulated, as follows: For neuroscience, five PET-images originating from the Coma Science Group at the University Hospital Liège (Steven Laureys) with rainbow color scales, and black backgrounds were used as a baseline. The images showed the brain of a healthy person, of a locked-in-person, of a person in minimally conscious state, of a person in vegetative state and a brain-dead person. The questionnaire included a more detailed description of each neurological state. For instance, for the locked-in state the instructions read: "the brain of the person has more or less normal consciousness state, but due to a neurological problem the person is unable to move and to communicate with the environment). All geographic stimuli were produced with ESRI's ArcMap10.2 geographical information system (GIS), using the raster image file format. This was done to assure that each pixel value in an image corresponds to a value in the color scale. The geographic displays represent land use categories in the Netherlands in 2000. This raster data set (with a spatial resolution of 500 meters per pixel) shows various land use categories, such as residential, industrial, recreation, and nature. The data set was provided by the LUMOS consortium, together with the Land Use Scanner software (see http://www. objectvision.nl/gallery/demos/land-use-scanner). We employed this software to simulate five

future land use scenarios for the Netherlands in 2030 (SPINlab, 2013, see https://spinlab.vu.nl/research/spatial-analysis-modelling/land-use-modelling/netherlands-2030/).

## Structure of the online questionnaire (also available as S1 File)

After the introduction page that contained information about the study, participants were asked to provide informed consent. Next, they answered general questions regarding age, gender and place of work. We also asked whether the participant suffered from any type of color vision deficiencies (i.e., no deficiency, weak red-green color deficiency, complete red-green color deficiency, yellow-blue color deficiency, complete color deficiency, that is, color blindness) in order to exclude them from the sample. No person was excluded based on this criterion. One person who had indicated a red-green color deficiency, had been randomly assigned the diverging blue-white-red color scale. We had chosen this colorblind safe scale for our study from an empirically validated color picker, specifically developed for maps (color-brewer2.org) as follows: https://colorbrewer2.org/#type=diverging&scheme=RdBu&n=8. We also checked this scale for colorblind safety using http://colororacle.org. We thus contend that this participant's judgments are not affected by this specific color vision deficiency.

Then, a first filter question was displayed where participants had to attribute themselves to one of the following three possibilities: Neuroimaging expert (education and/or work in neuroscience, neurology or a related field or professional work with data (neuroimages) emerging from those fields), geovisualization expert (education and/or work in geography, including cartography and geovisualization or a related field or professional work with geographic data (e.g., GIS, maps, etc. emerging from those fields) or none of the first two options (i.e., lay people). Neuroimaging experts then performed the trials containing neuroimages, followed by the trials containing the map images; for the geovisualization experts, the order was reversed (i.e., domain experts first answered questions with images with which they are more experienced). Lay people received the two blocks of images in a random sequence.

The image-portion of the experiment first started with a question regarding the attitude towards the visualization context, which all participants were asked to answer. In the "neuro-context", the statement to assess was "The state of brain death equals the death of a person" (4-point-Likert-skale from "completely agree" to "completely disagree" and "no opinion" as additional option), in the "geovis-context", the statement was "The climate change we experience now is caused by human activities". Then, the experiment involved two tasks: We first assessed participants' *trust* in the information conveyed by a visualization. Please note again our earlier definition of trust we used for the study relating to the validity of a conclusion that can be reached using a visualization. The second task assessed the impact of the visualization on data *interpretation*, namely ranking a state between the two extreme states. For example, in the neuroscience context, we presented participants with a map image of a healthy brain, accompanied by the statement that this image would show a healthy brain. Participants had to rate how much they trusted the image to support the statement on an 8-point Likert scale (see S1 File for the exact wording of the question). Subsequently, the image of a dead brain was displayed along with the statement that it visualized a dead brain. Once again, trust in the image supporting the statement had to be rated. Following that, participants were asked to rank three images of intermediate brain states in between images of two extreme states (a normal brain state at one end and brain death at the other end). Like the trust-rating task, a data scenario was provided for each of the three ranking tasks. One of eight conditions was randomly chosen; identical for both, the neuroimage trials and the geovis trials, respectively.

After the experiment, the experts answered detailed questions on their experience with creating visualizations, whereas the lay people only answered general questions with respect to

experiences with scientific visualizations (details see below). In a first block of questions, the persons were asked about their training (selection of various possibilities adapted to the domain), their current occupation, the number of years of experience in the domain, their professional relationship to image creation and use (various options), the size of the lab in which they work, years of experience of the lab in the domain, whether the lab produces images, and whether the participants considers themselves to be more consumers or producers of images in either domain (this was another filter question, see below). In this question block, participants were provided with the four color scales used in the experiment (see Fig 1A). They were asked which of the shown scales should be used to display the increase of a statistical parameter of any type in their field. We used a 4-point Likert scale for this, including not suitable, moderately suitable, suitable, and highly suitable). They were also asked whether one of the color scales should serve as a standard in their field.

Once the filter question was answered positively (i.e., the participant is indeed involved in producing images), additional questions were asked about how many images per year are produced, what kind of images, and for what purpose (e.g., in neuroimaging: clinical, etc.), which techniques were used for image data generation (e.g., in neuroimaging: PET, fMRI, etc.), which software is used for image generation (e.g., in neuroimaging: FSL, BrainVoyager etc.), how the software is used to enhance images, and whether and what kind of post-processing of images takes place, including the software used for post-processing.

After having completed the domain-specific questions, experts were also provided with five general questions related to their experience with images of the other domain, and thus identical to those of the lay people. For example, we asked them, whether they ever produced such types of maps themselves or collaborated with people producing such maps, whether they would regularly use such types of maps or whether they have a basic understanding of how such maps are produced, etc. (see S1 File for the exact wording of these questions). A general experience index was created by adding up each single item ticked by the participants.

Prior to data collection, the questionnaire was pretested first with nine domain experts in either neuroscience or geovis, and then in a sample of 27 participants of either domain, to test its usability. The response data of these participants have not been included in the main analysis. The questionnaire was subsequently adapted according to the feedback received. On average, neuroimaging experts took 22.5 minutes to complete the survey, geovisualization experts took 26.5 minutes, and the lay people needed a bit more than 10 minutes to complete the questionnaire. Their response time is shorter overall on average, because they were not asked to fill out the questions regarding expertise, except the five general questions about experience, as mentioned earlier.

### Ethics statement

The study has been approved following the procedures of the ethical evaluation of empirical studies at the Institute of Biomedical Ethics, University of Zurich (further information available at: https://www.ibme.uzh.ch/en/Biomedical-Ethics/Research/Ethics-Review-CEBES.html).

## Results

### Domain-specific response variability

We computed *trust variability* and *interpretation variability* and overall *response variability* (the sum of trust and interpretation variability) for each data context separately. For this, we averaged over all conditions in each, trust-ratings and interpretation rankings, and by calculating the absolute differences between the mean and the ratings for each individual participant.

For this analysis, we need to explain in more detail how we processed the response data. As mentioned earlier, the experiment consisted of two data *contexts* (i.e., neuro and geo), in which we had participants assess two *trust-related states* (e.g., in the neuro context: "normal brain" and "brain death") and three *interpretation-related states* (e.g., in the neuro context: locked-in, minimally conscious state, and vegetative state). Each state in each context has been presented to the participants in eight *conditions, that is*, four color scales and two image backgrounds). Let us take the neuro trust-related state "normal brain" as an example, as one of the eight conditions, denoted as *NB*1. We then calculated for all experts and all lay people who were confronted with this condition *NB*1 the overall mean of all trust ratings $\bar{T}_{NB1}$. This has been done for all other seven conditions in this state as well. Let $\bar{T}_{NB} = \{\bar{T}_{NB1} \ldots \bar{T}_{NB8}\}$ be the set of all means of this first trust state in the "neuro" context; let $\bar{T}_{BD} = \{\bar{T}_{BD1} \ldots \bar{T}_{BD8}\}$ be the set of all means of the second trust state (i.e., "brain death") and let $X_{NB}^{N} = \{x_{NB1}, \ldots x_{NB134}\}$ and $X_{BD}^{N} = \{x_{BD1}, \ldots x_{BD134}\}$, respectively, be the individual ratings achieved by all 134 neuroimaging experts *N*. The trust variability $T_n^N$ rating for all the neuroimaging experts in the data context "neuro" *n* is then calculated as:

$$T_n^N = \frac{1}{134} \left( \sum\nolimits_{X_{NB}} |\bar{T}_{NB} - X_{NB}^N| + \sum\nolimits_{X_{BD}} |\bar{T}_{BD} - X_{BD}^N| \right)$$

whereas $|\bar{T}_{NB} - X_{NB}^N|$ and $|\bar{T}_{BD} - X_{BD}^N|$, respectively, indicate that the absolute value of a data point of the set *X* is subtracted from the mean value $\bar{T}$ of the same context. The trust variability for all other groups (i.e., geovisualization experts and lay people) and in both contexts is calculated accordingly. We calculated *interpretation variability* similarly, by including all three interpretation-related states in our calculation. Finally, *overall variability* is the sum of the trust and interpretation variability.

Our first hypothesis was not confirmed (Fig 2). Instead, for the neuroscience data context, we found that response variability was significantly *greater* for neuroimaging experts compared to both geovis experts and lay people (mean score 7.34 vs. 5.87, and 5.73; p < .001, Mann-Whitney Test). This effect is due to trust variability (mean score: neuroimaging experts = 4.34, geovis experts = 3.16, lay people = 3.00; p < .001), and not to interpretation variability (mean score neuroimaging experts = 3.00, geovis experts = 2.70, lay people = 2.74). There are no significant group differences in the overall response variability in the geographic data context (mean score neuroimaging experts = 7.02, geovis experts = 7.14, lay people = 6.71; Fig 1B). Geovis experts show the highest interpretation variability, significantly higher than lay people (mean score neuroimaging experts = 3.09, geovis experts = 3.09, lay people = 2.81; p = .05 for geovis experts vs. lay people, Mann-Whitney Test). Display background (i.e., black vs. white) had no statistically significant influence on response variability in either data context.

Non-experts' responses seem to be unaffected by changes in color scales. One possibility for this unexpected result might be the exposure to domain-specific visualization standards. The use of a non-typical color scale could increase interpretation variability in domain experts [12], and thus reduce their trust ratings for that visualization in their own domain. We indeed found this pattern for neuroimaging experts. Specifically, when they were asked to assess brain scans depicted with the widely used rainbow color scale, in the case of PET scans [7], highest trust ratings and lowest interpretation variability for this scale in the neuro domain can be observed (Table 1). While trust in this color scale indeed remains high when assessing geographic data visualizations, it does not, however, decrease their data interpretation variability. Conversely, geovis experts were less influenced by domain-specific color standardization practices in their own domain. Paradoxically, their data interpretation variability show significantly greater differences across the assessed color scales in the neuroscience context, even though

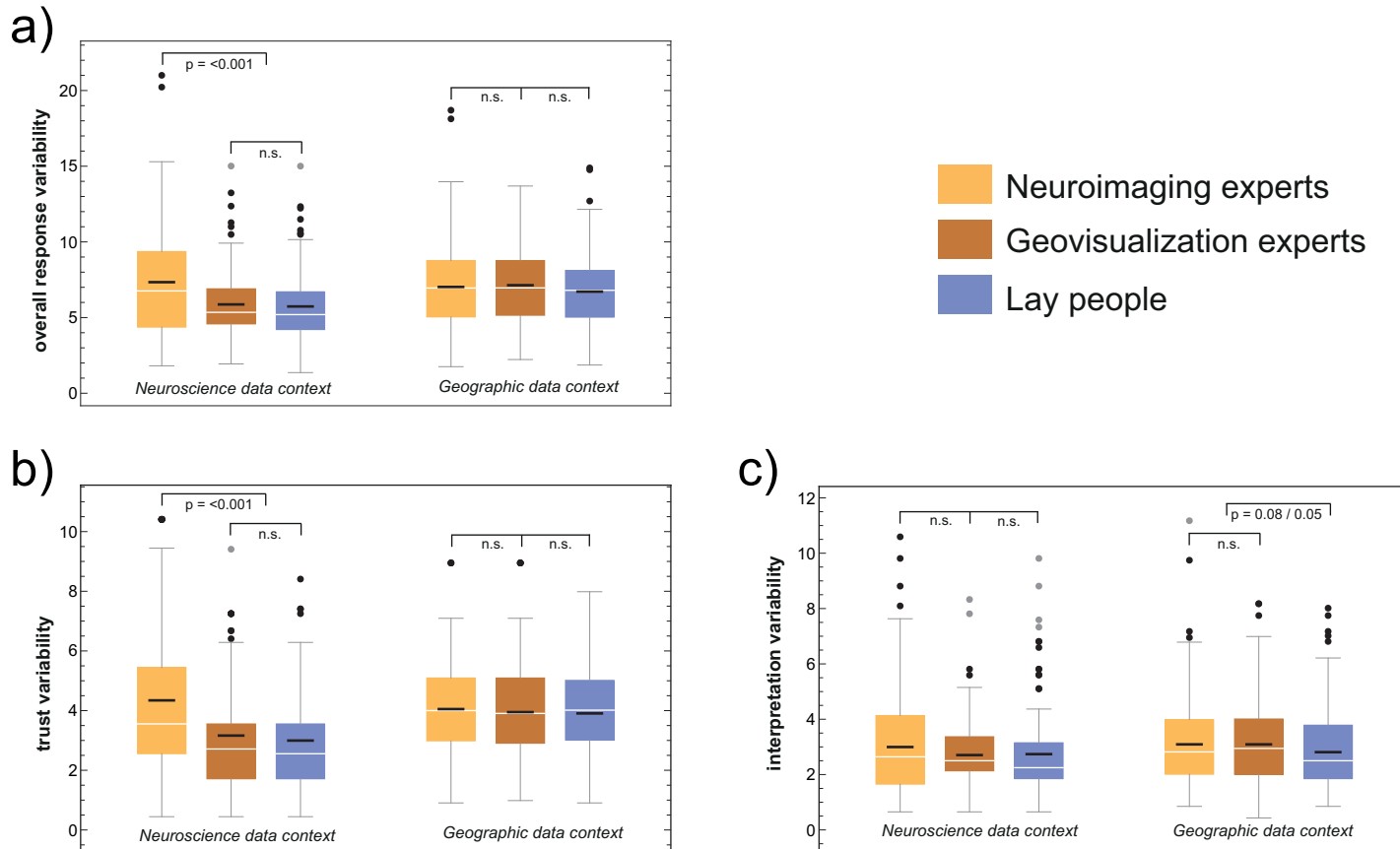

**Fig 2. Between-group comparisons of response variability.** Contrary to our hypothesis, overall response variability (higher values indicate greater variability) was highest for the neuroimaging experts in the neuroscience data context (a). This is mainly due to their greater trust variability (b). A tendency for experts' interpretation variability is only discernible in the geographic data context (c). Non-experts show lower variability in the neuroscience data context compared to the geographic context.

data encoding principles using color have a long history of standardization in cartography [12, 25].

## Factors potentially influencing response variability

We investigated several additional potential factors that could explain response variability across data domain context (see Methods section below). Firstly, we assessed whether experts' *experience in data visualization practice* might shed light on interpretation variability. No correlations between participants' visualization history and interpretation variability were found, suggesting that the latter is independent of expertise. Secondly, we investigated potential factors relating to common *practices in visualization display creation* across data contexts, to determine whether one of these factors might affect overall data interpretation variability. Again, none of the factors (e.g., reference to technical literature, lab rules, or software used, etc.) significantly affected participants' response variability. Finally, we assessed the potential influence of *attitudes towards visualization standardization* and the *credibility of data contexts*. Although we found significant group differences (e.g., as predicted by cartographic design theory, that is, geovis experts considered the rainbow scale to be least trustworthy), attitudes towards color scales were uncorrelated with response variability.

**Table 1. Results across color scales.**

| Group | Task | Color Scale (group size neuro/geo) | Data context [Mean (SD)] | |
|---|---|---|---|---|
| | | | neuro | geo |
| **Neuroimaging experts** | **Trust rating values** | Rainbow (n = 54 / n = 70) | 6.17[a] (1.98) | 5.91[b] (2.22) |
| | | Heated-body (n = 78 / n = 64) | 5.15 (2.54) | 5.20 (2.26) |
| | | Univariate color intensity (n = 70 / n = 66) | 4.76 (2.70) | 5.17 (2.12) |
| | | Bivariate blue-white-red (n = 66 / n = 68) | 4.85 (2.70) | 4.87 (2.38) |
| | **Interpretation variability** | Rainbow | 5.67[c] (2.59) | 6.53 (2.14) |
| | | Heated-body (i.e., "light-is-more") | 7.66 (2.76) | 6.75 (2.01) |
| | | Univariate color intensity (i.e., "dark-is-more") | 7.95 (2.75) | 6.49 (2.29) |
| | | Bivariate blue-white-red | 7.97 (2.0) | 7.55 (3.65) |
| **Geovis experts** | **Trust rating values** | Rainbow (n = 58 / n = 72) | 6.29 (1.89) | 4.99 (2.36) |
| | | Heated-body (n = 64 / n = 76) | 6.72[d] (1.99) | 5.28 (2.20) |
| | | Univariate color intensity (n = 78 / n = 54) | 6.44 (1.80) | 4.63 (2.16) |
| | | Bivariate blue-white-red (n = 68 / n = 66) | 6.46 (2.29) | 4.92 (2.37) |
| | **Interpretation variability** | Rainbow | 5.41 (2.45) | 7.27 (2.57) |
| | | Heated-body | 8.93[e] (2.38) | 6.70 (2.26) |
| | | Univariate color intensity | 5.34 (1.87) | 6.92 2.27) |
| | | Bivariate blue-white-red | 6.16 (2.95) | 6.67 (2.38) |

The table shows trust-ratings (higher values = greater trust) and mean interpretation variability (higher values = greater variability; group sizes are equal to trust ratings), and standard deviations (below the mean) across color scales, data domain context (i.e., neuroscience vs. geography), and domain expertise (i.e., neuroimaging vs. geovis experts). (p-values refer to the comparison between the scale indicated in the row and the other three scales: a: .02–.003; b: .05–.004; c: .004–.001; d: .03 vs. rainbow; e: < .001 in all cases).

Our data allow evaluating whether rated general suitability for a given scale in the respective application domain (neuroimaging versus geovisualization; see methods part) would determine *trust* in data represented with that scale. Fig 3 shows the result of this comparison for neuroimaging and geovis experts. We find that geovis experts' suitability ratings match the visualization principles recommended by long-standing cartographic design theory. That is, the color scale displaying decreasing data magnitudes with lighter color values (i.e., less intensity) obtained the highest suitability ratings. Least preferred was the rainbow scale. Similarly, neuroimaging experts show the highest suitability ratings for the heated body scale, the most commonly used scale in functional neuroimaging.

However, when calculating the difference between suitability ratings and actual trust ratings expressed in the experiments (see methods for details), we again find a surprising pattern. Participants tend to exhibit much more trust in the rainbow scale than would have been predicted by their suitability ratings. This discrepancy is strongest for the geovis experts, who are typically trained to not use the rainbow scale for quantitative data encodings. Likewise, and similarly puzzling, is the result for the heated body scale. As expected, neuroimaging experts exhibit greatest suitability ratings for this particular color scale, widely used in their community. However, their trust in this scale is rather modest. A different trend can be seen for geovis experts. They seem to trust the heated body color scale most, although there are no significant differences compared to the other scales, and only a tendency when comparing with univariate color intensity change (p = .08), even though it is almost never used for geographic datasets. This trust rating is surprising, because the heated body scale is the direct opposite of the well-established cartographic design principle to assign darker color values (i.e., color intensities) to convey greater data magnitudes (i.e., the decreasing color value scale in Fig 1). This might explain why geovis experts' suitability ratings are second lowest for the heated body scale (least

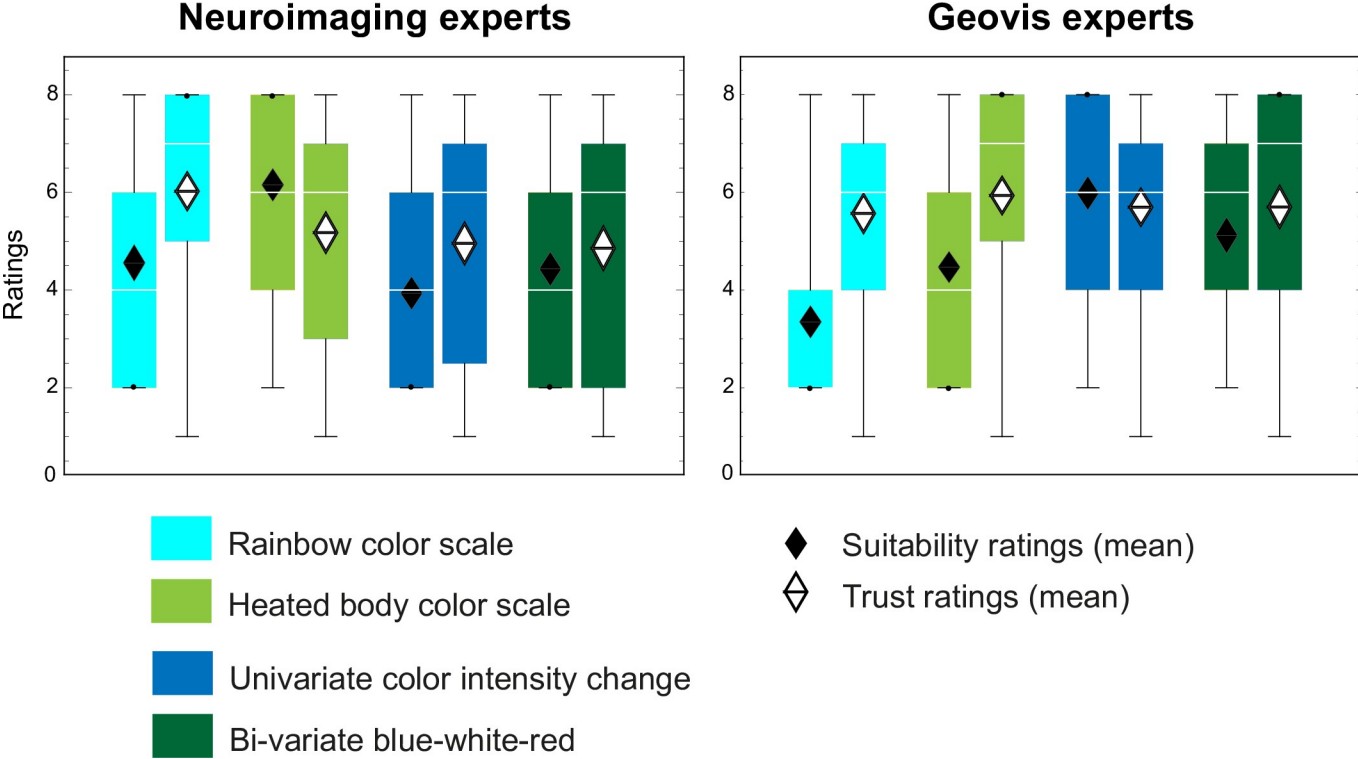

**Fig 3. Comparing suitability with trust ratings.** General suitability ratings for color scales to be used in a domain (i.e., neuroimaging or geovisualization) do not align with trust ratings collected during the experiment. This is particularly true for the rainbow scale. This effect is strongest for the geovis experts, trained not to use the rainbow scale in the illustration of univariate data in a progression [13, 14].

preferred was the rainbow color scale). For the decreasing color value (light-to-dark) and the bi-variate (cold-to-warm) color scales, trust ratings and self-reported suitability ratings were aligned more closely.

## Other variables

None of the many variables potentially related to visualization expertise correlated with overall data interpretation variability; with two notable exceptions (see S1 Table). Neuroimaging experts who claim to more strongly rely on their "intuition" when choosing color scales for their visualization needs tend to show less variability (overall correlation: -0.12, p = .08), but only for the relatively unfamiliar geographic data context (-0.20, p = 0.03). In contrast, geovis experts, who more strongly rely on cartographic design theory (-0.12, p = .05) tend to exhibit less response variability; but only for the geographic data context they are especially familiar with (-0.16, p = .07).

We also assessed whether experts' *experience in data visualization practice* might shed light on interpretation variability. For this, two complementary interpretations are possible. On the one hand, when experts are shown data with a well-known color encoding principle, visualization experience supports the appropriate decoding of depicted data. An uncommon color encoding would then *increase* interpretation variability, and one would expect a positive correlation between visualization expertise and interpretation variability in that particular data context. On the other hand, visualization experience in a data domain could lead to an increased awareness of problematic color scales, limiting possible bias for the preferred scale. Consequently, one would expect interpretation variability to *decrease* with increasing expertise (i.e.,

a negative correlation). However, no correlations between participants' visualization history and interpretation variability were found, suggesting that the latter is independent of expertise.

We further investigated potential factors that relate to common *practices in visualization display creation* across data contexts, as to determine whether one of these factors might affect overall data interpretation variability (see methods for details). None of these factors affect participants' response variability. We only found that, overall, geovis experts rely more often on their intuition when determining color encodings for visualization (N: 2.58 vs. G: 2.88, p = .03), whereas neuroimaging experts rely more often on imaging software default settings (2.84 vs. 2.04, p < .001) and lab internal conventions (2.04 vs. 1.80, p = .04). Geovis experts also more often change predefined color scales (N: 2.04 vs. G: 2.38, p = 0.01), or even modify visual displays (i.e., delete or emphasize parts of a visualization in a graphic display) (N: 1.42 vs. G: 2.28, p < .001), whereas neuroimaging experts more often rearrange visualizations in a graphic display (N: 2.87 vs. G: 2.45, p = .002) or add arrows or other markers to highlight specific aspects in a visualization (N: 2.79 vs. G: 2.23, p < .001).

Finally, we systematically assessed the potential influence of *attitudes towards visualization standardization* and the *credibility of data contexts*. Here we find significant group differences: geovis experts are significantly more critical about brain death to indicate the death of a person (N: 1.75 vs. G: 1.99; p = .02) compared to neuroimaging experts. As predicted by cartographic design theory, geovis experts consider the rainbow scale to be least trustworthy (N: 6.02 vs. G: 5.57, p = .05). This lowest ranking is followed in increasing order by the more common blue-red (N: 4.86 vs. G: 5.70, p = .002), the green–light yellow (N: 4.96 vs. G: 5.70, p = .01), and at the top of the list, and most surprisingly, the least known heated body scale (N: 5.18 vs. G: 5.94, p = .004) to be most trustworthy compared to neuroimaging experts. Overall, however, we found that participants' attitudes, both with respect to color scales and the encountered data contexts, do not correlate with their response variability. Some small but significant (Spearman's rank) correlations were found when assessing the trust ratings alone. Both neuroimaging (-0.18, p = .04) and geovisualization experts (-0.18, p = .04) were less likely to agree that the image showing a "dead brain" supports the claim that the brain is actually dead. We did not find this for lay people. For the climate change paradigm, a negative correlation (-.011, p < .01) has only been found when combining the data of all three groups, but not for any of the tested groups individually.

## Discussion

The current study was squarely embedded within a larger research program to uncover mechanisms and standards in data visualization, specifically in neuroimaging [7]. It has important implications for data visualization practice beyond the biological and geographical sciences, spanning the social sciences, the humanities, and the engineering sciences. In our previous studies, we found an increasing popularity of the heated body scale to denote increasing neuronal activity (or increasing statistical significance) and the use of "cold" colors for a decrease in activity (or decreasing statistical significance). However, the degree of standardization still limps far behind the one reached in, for instance, cartography and geographical information visualization, which is why we used this application domain for comparison. For example, in cartography, the selection of color shades parallel to the progression of data values–the higher the data magnitudes, the darker the color shade–has emerged as one of the few standards by the end of the 19th century [26]. Since then, color progressions are by convention also used in cartography to depict quantitative data sets, for example in a thematic map, when using data at the ordinal (i.e., high-medium-low incidents of crime), interval (i.e., day temperatures in degrees Celsius), or ratio level (i.e., number of inhabitants per country) of measurement. For

these kinds of data sets single-hue, bi-polar hue, complementary hue, and value progressions (i.e., light-to-dark), and more recently two- and more variable color progressions and multi-variate blends are commonly used in statistical maps. Many of such cartographic conventions have been tested by time and eventually found to be successful because of their commonly accepted use [18]. More recently, cartographic design conventions have been empirically assessed and found to be working as predicted [4, 14, 18], including the principles for the systematic application of color in maps [27], and statistical data representations [2]. Neuroimaging and other data-driven scientific domains could take these long-standing and successful mapping principles and data visualization conventions as a starting point.

Our hypotheses a) to d) were based on these considerations; in particular hypothesis d), where we expected that geographic information visualization experts would show greater awareness on sound design principles compared to neuroimaging experts. However, we found that (a) domain experts do not have the lowest response variability in both the trust and the interpretation task, and that (b) response variability is unaffected by professional experience in data visualization. This is a somewhat surprising finding, as our previous research indicated some trends of standardization in neuroimaging both over time and on the level of institutional sub-units [7]. Our results also suggest that (c) potentially ideologically charged opinions related to the data context (in our case related to brain death as a death criterion or the degree to which human activities cause climate change) do not influence overall response variability when assessing visualizations on specific phenomena related to the two domains of application. This finding does not contradict found correlations of mistrust in a visualization that would support a given claim (i.e., regarding brain death or a "collapsed ecosystem") and the general belief in the concept of brain death or human-caused climate change. The point here is that the overall assessment of the various states such as, "locked in" etc. in the neuroimaging paradigm, for instance, remains unaffected in both application domains. This finding, in line with our hypothesis, is important, as it diminishes fears of "strategic" design choices to convey preconceptions in data; a known warning in information design and data visualization [28]. Finally, most surprisingly, we find that (d) although geographic visualization experts are aware of the limitations of the rainbow color scheme and trained not to use it, this expertise is not reflected in their trust ratings for data displays using the rainbow scale. A dissonance of what people believe, prefer, or feel, and how they actually perform, and/or what is best for them is not an uncommon finding in psychological research concerning many contexts–and geovisualization is no exception [29]. A potential difference of trusting the content of a visualization, compared to making effective and efficient decisions with that visualization might, however, be especially problematic in a geographic context using the rainbow scheme when public policy decisions need to be made such as for a global climate change context [12, 22]. We wish to emphasize here again that our research intention was to investigate *how strongly* participants would trust an expert's claim supported with given visualized data. This trust would be independent of the trust in the visualized data per se. We chose this assessment because we wished to avoid any domain expert participants feeling the need to assess the appropriateness of a particular display method. So, for example, we intended to avoid neuroscience experts judging, say, PET, to be an appropriate display method for assessing disorders of consciousness. While we aimed for increased ecological validity with our approach, we acknowledge that the latter, that is, distrust in the data as such, could have had an effect in participants' responses too.

As with all empirical research, our results must be understood in the context of specific study limitations. Firstly, to avoid making the experiment even longer, we assessed color vision deficiencies with self-reports, instead of screening participants' potential color vision deficiencies with standardized test instruments. Therefore, participants unaware of their color vision deficiencies may have participated in our study. However, only visualizations with the rainbow

color scheme could have been potentially affected by this limitation. Secondly, we assessed lay people's general experience with visualizations using identical questions also given to the expert participants, despite the fact that they did not have domain expertise. The reason for this was to increase comparability between experts and lay people. However, we did not include questions to assess lay people's affinity with data and/or scientific visualization in general. Third, because of the popularity and ubiquity of GPS-enabled mobile maps on people's smart phones, people's hobbies and mobility activities in their leisure time may require the use of geographic information (i.e., hiking, sailing, travels, road trips, etc.) it is conceivable that lay people may be more familiar with the use of cartographic maps in general, compared to brain maps [30]. Comparing the geo and neuro image types in our collected data, we find that for both, lay people and experts in the domain for which they do not have expertise, exhibit a significantly lower general experience for neuroimaging maps (lay: 0.72, geo: 0.69; difference between these two groups is not significant) compared to geographic maps (lay: 0.99, neuro: 0.90; difference between these two groups is not significant). This may result in a lower response variability by lay people for the neuroscience images. One could argue that lay people might not feel experienced enough to question neuro expert authority or to have strong feelings or opinions about this domain. Their response variability may be compressed as a consequence. This could be in contrast to the geovis experts, who might feel to have less working experience with neuro images specifically, but would at least have familiarity and confidence in working with visualized data in general.

In summary, our results suggest that 3 out of 4 of our theory-driven hypotheses, based on the extensive review of respective prior research [7], were not supported by our empirical results. This is in spite of the fact that they are also based on unequivocal "common sense" opinion and practice in the data visualization literature [2, 9–13]. There are indications that the pronounced effect of trust variability in case of neuroimaging experts results from the fact that the rainbow scale is quasi-standard in PET imaging. However, the missing correlation with imaging experience puts this interpretation into question. While in the geovisualization literature the critical discussion of the rainbow scheme is documented [13, 14]. Nevertheless, it has become a quasi-standard for visualizing climate data and remotely sensed imagery of various content, because this color scheme has been for a long time one of the default settings in respective software [19, 20]. Recent studies confirm the problematic aspect of the rainbow scale such as the perceptual discretization of the continuous data space on maps, caused by changing hues in the rainbow color scale sequence, and the finding that different datasets create unpredictable variations in the perceived hue bands [31]. A proper understanding of the mechanisms that drive these phenomena remains unclear. Furthermore, the discrepancy between knowledge of sound visualization design standards and the lack of an appropriate application, points to the difficulty to establish such standards in practice.

Given that, in a big data era, visualization is expected to play an increasingly significant and growing role in the data-driven sciences for data analysis and communication of results, increasing awareness for sound design principles in data visualization becomes an urgent matter [32]. To reach this goal, it is key that visualization experts closely cooperate with cognitive scientists, familiar with the mapping of colors to concepts. Some factors that determine this mapping have been anticipated intuitively [13], but may still have to be refined empirically, to optimize the most automatic way from color to meaning [12, 13].

## Supporting information

**S1 File.**
(PDF)

**S1 Table.**
(DOCX)

# Acknowledgments

The authors thank Gianluca Boo (Research Fellow at WorldPop Project at the University of Southampton) for producing the images used in the study. We also thank Steven Laureys (Coma Science Group Chair and Clinical Professor of Neurology at the University Hospital Liège, Belgium) for the raw brain images that have been used for this study.

# Author Contributions

**Conceptualization:** Markus Christen, Peter Brugger, Sara Irina Fabrikant.

**Data curation:** Markus Christen.

**Formal analysis:** Markus Christen.

**Methodology:** Markus Christen, Peter Brugger.

**Software:** Sara Irina Fabrikant.

**Supervision:** Peter Brugger, Sara Irina Fabrikant.

**Writing – original draft:** Markus Christen, Peter Brugger, Sara Irina Fabrikant.

**Writing – review & editing:** Peter Brugger, Sara Irina Fabrikant.

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
