## [Decision Letter · Decision Letter 0]

13 Oct 2020

PONE-D-20-23193

Susceptibility of domain experts to color manipulation indicate a need for design principles in data visualization

PLOS ONE

Dear Dr. Christen,

Thank you for submitting your manuscript to PLOS ONE. Your manuscript has been reviewed by two external experts. Based on their recommendations, we invite you to submit a revised version of the manuscript that addresses the points raised during the review process.

As you will see in the reviews below, both reviewers believe that this is an interesting study, but the also pose several concerns. In particular, both reviewers indicate that numerous important details are missing about the study methods. In addition, one of the reviewers is concerned that there may be alternative explanations of your results that should be addressed.

We look forward to receiving your revised manuscript.

Kind regards,

Ronald van den Berg

Academic Editor

PLOS ONE

Journal Requirements:

Additional Editor Comments (if provided):

Reviewers' comments:

Reviewer's Responses to Questions

**Comments to the Author**

1. Is the manuscript technically sound, and do the data support the conclusions?

Reviewer #1: Yes

Reviewer #2: Partly

2. Has the statistical analysis been performed appropriately and rigorously? 

Reviewer #1: Yes

Reviewer #2: Yes

3. Have the authors made all data underlying the findings in their manuscript fully available?

Reviewer #1: Yes

Reviewer #2: Yes

4. Is the manuscript presented in an intelligible fashion and written in standard English?

Reviewer #1: Yes

Reviewer #2: Yes

5. Review Comments to the Author

Reviewer #1: This manuscript describes a study with three populations—one in neuroscience, the second in geovisualization, and a third of lay users---that measures subjective trust and interpretation of heatmap-style visualizations. The concept explored is intriguing and relevant: how does expertise influence colormap preference and use. Further, the paper explores an extensive range of relationships between visualization expertise and the collected measures. Overall, the study is interesting and has significant potential for challenging assertions around design knowledge and visualization use. However, there are several missing details and potential biases that may influence the results that should be addressed before the paper is ready for publication.

- Missing Details:

The description of the study data is reasonably clear. The use of real-world datasets and conventions from the tested fields is a significant strength of the work. However, there are several missing details that may help provide stronger context for interpreting the reported results.

While there are examples of how trust data collected, there is no description of how trust was defined to participants. Different conceptualizations of trust may influence how participants report trust. For example, how well do you trust yourself to use this visualization effectively seems like it may be more influenced by color choices than how well do you trust the data in this visualization to help you draw conclusions. While the latter framing is more ecologically valid (and seems closer to the likely framing based on evidence in the paper), it also leads to a bias whereby people may be willing to trust the visualization they’re using for a given task even though they know it is suboptimal (i.e., to report it as “good enough”). This is touched on in the discussion, but having more detail as to how trust was conceptualized (or interpreted) would significantly strengthen and contextualize the discussions of trust.

I appreciate the thoughtful considerations of CVDs. However, was any sort of CVD screening carried out? Many people do not know they have any sort of CVD until adulthood, making self-reporting challenging and at times inaccurate. This is a minor limitation as online CVD tests are imperfect due to a number of factors, like monitor calibration, but it is worth explicitly mentioning.

The manuscript notes that five common demographic questions were provided to all participants. However, no information about those five questions is provided. Reading the results section, it appears that these questions were used to draw conclusions about the data; however, having a summary of these questions would be helpful for understanding the study. For example, did any common questions assess lay participants’ comfort with data generally or experience with design?

For the statistical reporting, it would be helpful to have some sense of the precision of the provided means (e.g., standard deviations or confidence intervals) reported along side the means. It would also be helpful to explicitly define how variability is computed (e.g., standard deviation, variance, etc. and within subjects, over the same set of images, etc?). These details would significantly clarify the significance of the observed effects (or lack of effect in some cases).

- Potential Confounding Biases:

I appreciate the thorough discussion of potential confounding factors. The thoughtful analysis of factors around domain and design conventions is a nice addition to the paper. Further, the lack of evidence that expertise has any influence on trust and interpretation raises fascinating questions about the influence of design knowledge and thinking in visualization. However, two potential confounds that may explain at least some of the observations arise from the design of the study itself rather than from the colormaps or visualization experience, namely familiarity/risk and confirmation bias. These concerns seem problematic given the reliance on a lack of result used to drive many conclusions in the paper.

First, we expect that people are more familiar and comfortable with maps and space than with medical images. Medical images lead to both higher risk decisions and to less ability to personalize the data (see Peck et al, 2019), both of which are correlated with lower trust and confidence in an image and may provide an alternative explanation for the observed behaviors. This may lead to lower lay variability in the neuroscience images: they don’t feel they have the experience or authority to levy strong feelings or opinions and their response scales may be compressed as a consequence whereas geovis experts at least have familiarity and confidence in working with visualized data. It may also explain the higher criticality around brain death observed for neuroimaging experts over other populations.

Second, the question used in the geovis context appears like one that would be highly subject to confirmation bias. Most lay people have strong opinions on whether or not climate change is caused by human activities. As a result, confirmation bias may have a significant impact on responses to geovis questions compared to the neuroimaging questions, which focus on specific examples. This appears to play out in the high overall scores (mean of above 7 out of 8 for experts populations may indicate ceiling effects).

The discussion appears to point to that would invalidate this bias, “Our results also suggest that (c) potentially ideologically charged opinions related to the data context (in our case related to brain death as a death criterion or the degree to which human activities cause climate change) do not influence response variability, which is in line with our hypothesis.” However, it isn’t clear from the discussion of results (which primarily focus on expertise as relates to visualization) where the direct evidence of this division arises from. It would be helpful to the reader to more explicitly bridge this claim with the data itself.

While these alternative explanations do not invalidate the results, they do suggest limitations in considering the magnitude and outcomes of the provided results and for guiding future study. A revised manuscript should include an evidence-based discussion of how the general problem framing may invite specific kinds of biases and/or how those biases are mitigated by the infrastructure itself.

In summary, the research presents an interesting line of inquiry into understanding the role of expertise in assessing visualizations with various color encodings. While the study offers a number of interesting observations, there are several missing details and potential alternative explanations that should be addressed in the manuscript before it is ready for publication.

- Grammatical Issues:

“each of the tree ranking tasks”—three

“that that (b) response”—extra “that”

Reviewer #2: This paper presents a study whose focus about color usage in visualizations has the potential to offer value to research and practitioner communities. It posits interesting hypotheses that certainly are well-posed and of interest. Some aspects of the methods, results, and presentation need some attention, though - these few, probably correctable, problems with the manuscript at present make it not yet suitable for publication.

In my view, the conclusion here that certain experts exhibited a higher-than-predicted trust in the rainbow scale is in agreement with some other reported studies and with some small-sample informal tests by colleagues. Those other reports have been to the surprise of some who have had discussions about them. This work here I think adds to the evidence that we are going to have to have further, deeper analysis and care in our views of the rainbow scale. Thus, the work could both increase our current knowledge status and lead to future increases, too. That is a nice plus, counseling for a revised, improved manuscript that PLOS ONE might later accept.

Some more detailed points about the areas needing attention follow.

1. Of the 134 neuro email addresses, is there a possibility that some of these belong to a common person? That is, could some individual(s) have been represented

multiple times in the responses, or were steps taken to somehow remove responses from an individual with multiple email addresses?

2. Of the 574 geographer email addresses, is there a possibility that some of these belong to a common person? That is, could some individuals have been represented multiple times in the responses, or were steps taken to somehow remove responses from an individual with multiple email addresses? The problem seems potentially more severe here (than with the neuro people) as there were also some addresses from ICA here.

3. I wonder if some "master experts" who were contacted handed off their response to someone else. One way to test that would be to determine if any of the inputs came from email addresses different than those who were emailed (at least for the neuro people).

4. How much were the Mechanical Turk people paid? How much were the domain experts paid?

5. The term "locked-in" in "Image Creation" is not well-known; a different term is needed.

6. There is a typo in "Image Creation", Sentence 3: "graphic" -> "geographic".

7. There are two grammar errors in "Image Creation": Sentence 6, "such as, residential" should be ", such as residential"; Sentence 7, "set is" should be "set was"

8. In "Structure of the online...", it should be mentioned if any of the Mechanical Turk participants were classified as experts? In addition, if any from the neuro or geography groups were classified as lay (or if none were), that should be mentioned.

9. In "Structure of the online...", Paragraph 3, "likert" should be "Likert".

10. In "Structure of the online....", Paragraph 3, it should be made clear if the lay people answered all these questions, also.

11. In "Structure of the online....", Paragraph 4, "more a consumer or a producer" should be "more consumers or producers".

12. In "Structure of the online....", Paragraph 4, what is "of main text" referring to? Is there another Figure 1 in another component that was not submitted?

13. In "Structure of the online....", Paragraph 6, Sentence 2, this point should also be made earlier--just before Paragraph 2 of "Structure of the online..." (or perhaps only made there).

14. Who were the 27 participants that tested questionnaire usability--were any of them the same ones that took part in the other parts of the study? Did any of the 27 that tested usability drop out due to the length of the survey completion?

15. There is a sentence frag. just above Table 1 ("Specifically....").

16. Figure 3 seems misleading as it aggregates the Table 1 information without regard to context. I'm not convinced of the conclusions related to such aggregated material...domain experts, especially if they have a long history in their domains, may tend to become very domain-centric and thus perhaps more likely to use their own training as they approach another domain, which may be a serious error for the other domain. I think the Figure 3 material should be de-aggregated, with the study then considering each context for each expert set, without considering "geo" as a whole, for example. Alternately, the focus could be just neuro experts on neuro data and geo experts just on geo data.

17. In "Factors potentially," the focus suddenly shifts to "preference." I don't see where that comes from. Moreover, the Figure 3 material is suitability versus trust, NOT preference versus trust. There is a difference between something being suitable and something being preferred. This part of the paper needs some re-thought and re-work.

18. The current Figure 3 is not convincing to me that "geovis experts seem to trust the heated body color scale most" (statement just before the "Other variables" section) - the trust levels are pretty close for all. Either a statistical showing there is needed or the statement needs to be removed.

19. Table 2 has a low information-to-space ratio. The table should be reformatted in some way to be made much more compact - nearly every entry there is "ns" but the table takes up about 5% of the entire page length of the paper.

20. p. 19, "Our hypotheses" paragraph, line 5, "that that" should be "that".

21. Ref. 2 is incomplete.

Lastly, a final positive remark:

* Next-to-last paragraph last sentence makes a very good point!

6. PLOS authors have the option to publish the peer review history of their article (what does this mean?). If published, this will include your full peer review and any attached files.

Reviewer #1: No

Reviewer #2: No

---

## [Author Response · Author response to Decision Letter 0]

26 Nov 2020

Susceptibility of domain experts to color manipulation indicate a need for design principles in data visualization – Response to Reviewers

We thank the reviewers for their detailed and constructive review. Below we respond to revision suggestions by reviewers in turn. All respective changes in the revised manuscript are marked in yellow in the document “Revised Manuscript with Track Changes”. We did not highlight linguistic or grammatical corrections for better readability.

Editorial remarks

We have checked the requirements and made changes accordingly. In particular, we updated the reference style.

Please include additional information regarding the survey or questionnaire used in the study and ensure that you have provided sufficient details that others could replicate the analyses. For instance, if you developed a questionnaire as part of this study and it is not under a copyright more restrictive than CC-BY, please include a copy, in both the original language and English, as Supporting Information.

We made the questionnaire available as Supporting Information.

We note that you have stated that you will provide repository information for your data at acceptance. Should your manuscript be accepted for publication, we will hold it until you provide the relevant accession numbers or DOIs necessary to access your data.

We have uploaded the dataset on Zenodo; the DOI is: 10.5281/zenodo.4284817

Reviewer 1

R1-1 While there are examples of how trust data collected, there is no description of how trust was defined to participants. Different conceptualizations of trust may influence how participants report trust. For example, how well do you trust yourself to use this visualization effectively seems like it may be more influenced by color choices than how well do you trust the data in this visualization to help you draw conclusions. While the latter framing is more ecologically valid (and seems closer to the likely framing based on evidence in the paper), it also leads to a bias whereby people may be willing to trust the visualization they’re using for a given task even though they know it is suboptimal (i.e., to report it as “good enough”). This is touched on in the discussion, but having more detail as to how trust was conceptualized (or interpreted) would significantly strengthen and contextualize the discussions of trust.

We fully agree with this reviewer that the “trust” concept is complex and could involve various aspects such as trusting the data as such or trusting whether a certain visualization correctly translates the content of the data. We have clarified on page 4 that in our survey, the notion of trust concerns whether participants believe the data presented in a certain type of visualization supported an empirical claim by a domain expert. We refer to the exact wording of the trust-related questions on page 9. The original survey document is now added as supplementary information. We added a short discussion on the various ways how trust could be understood on page 22.

R1-2 I appreciate the thoughtful considerations of CVDs. However, was any sort of CVD screening carried out? Many people do not know they have any sort of CVD until adulthood, making self-reporting challenging and at times inaccurate. This is a minor limitation as online CVD tests are imperfect due to a number of factors, like monitor calibration, but it is worth explicitly mentioning.

On page 22, we added respective information that no CVD screening had been performed and why this is the case. We also further explain why we believe that the potential effect of this limitation is unlikely to have affected the main results of our study.

R1-3 The manuscript notes that five common demographic questions were provided to all participants. However, no information about those five questions is provided. Reading the results section, it appears that these questions were used to draw conclusions about the data; however, having a summary of these questions would be helpful for understanding the study. For example, did any common questions assess lay participants’ comfort with data generally or experience with design?

On page 10-11 we have added examples of the survey questions used to assess “general experience” and we refer to supplementary information that includes the entire questionnaire. These questions aim to assess experience with the design of such maps. We did not ask lay people about their general comfort with data. This has been added as a limitation on page 22. We thank the reviewer for helping us clarify this point.

R1-4 For the statistical reporting, it would be helpful to have some sense of the precision of the provided means (e.g., standard deviations or confidence intervals) reported along side the means. It would also be helpful to explicitly define how variability is computed (e.g., standard deviation, variance, etc. and within subjects, over the same set of images, etc?). These details would significantly clarify the significance of the observed effects (or lack of effect in some cases).

In Table 1, we added the standard deviations to all computed values. On page 11-12, we added detailed information on how trust variability, interpretation variability, and overall response variability have been computed.

Potential Confounding Biases: I appreciate the thorough discussion of potential confounding factors. The thoughtful analysis of factors around domain and design conventions is a nice addition to the paper. Further, the lack of evidence that expertise has any influence on trust and interpretation raises fascinating questions about the influence of design knowledge and thinking in visualization. However, two potential confounds that may explain at least some of the observations arise from the design of the study itself rather than from the colormaps or visualization experience, namely familiarity/risk and confirmation bias. These concerns seem problematic given the reliance on a lack of result used to drive many conclusions in the paper.

R1-5 First, we expect that people are more familiar and comfortable with maps and space than with medical images. Medical images lead to both higher risk decisions and to less ability to personalize the data (see Peck et al, 2019), both of which are correlated with lower trust and confidence in an image and may provide an alternative explanation for the observed behaviors. This may lead to lower lay variability in the neuroscience images: they don’t feel they have the experience or authority to levy strong feelings or opinions and their response scales may be compressed as a consequence whereas geovis experts at least have familiarity and confidence in working with visualized data. It may also explain the higher criticality around brain death observed for neuroimaging experts over other populations.

We thank the reviewer for this observation. Our data indeed allows estimating differences in familiarity and we do find significant differences with respect to the general experience with geo-images compared to neuro-images. Both lay people as well as geography experts have lower general expertise index values (a term that now is explained on page 10) for neuro-images compared to geo-images and these values are comparable for lay people and geography experts. We now discuss this potential confounding factor on page 22 by also citing the article of Peck et al. 2019.

R1-6 Second, the question used in the geovis context appears like one that would be highly subject to confirmation bias. Most lay people have strong opinions on whether or not climate change is caused by human activities. As a result, confirmation bias may have a significant impact on responses to geovis questions compared to the neuroimaging questions, which focus on specific examples. This appears to play out in the high overall scores (mean of above 7 out of 8 for experts populations may indicate ceiling effects).

Thank you for raising this point. We did indeed assess to what extent participants believe in a brain death claim or landscape death claim, as indeed both claims can and are being contested (i.e., see ongoing public debates on climate change and/or pro-live discussions). As outlined on page 20, however, differences with respect to these claims do not correlate with the recorded response variability, which speaks against a potential confirmation bias. Furthermore, we are unsure to what the remark regarding “ceiling effect” refers. In the absolute trust ratings (Table 2) no ceiling effects appear; the interpretation variability data is calculated differently (as not outlined) and no direct relation to the scale used (Likert scale 1-(9 can be made here. We therefore believe that this is not a general point of concern.

R1-7 The discussion appears to point to that would invalidate this bias, “Our results also suggest that (c) potentially ideologically charged opinions related to the data context (in our case related to brain death as a death criterion or the degree to which human activities cause climate change) do not influence response variability, which is in line with our hypothesis.” However, it isn’t clear from the discussion of results (which primarily focus on expertise as relates to visualization) where the direct evidence of this division arises from. It would be helpful to the reader to more explicitly bridge this claim with the data itself.

We have added more detail on this point, both in the result section on page 19 and in the discussion section on page 22. In particular, we clarified that some small (but indeed significant) correlations between trust and opinion exist. However, these do not play out in overall response variability.

R1-8 While these alternative explanations do not invalidate the results, they do suggest limitations in considering the magnitude and outcomes of the provided results and for guiding future study. A revised manuscript should include an evidence-based discussion of how the general problem framing may invite specific kinds of biases and/or how those biases are mitigated by the infrastructure itself.

As outlined above, we added our responses to these relevant points to the discussion section. We thank the reviewer for these opportunities for clarification. We thus added a paragraph on page 22 that summaries the limitations of the study.

Grammatical Issues:

“each of the tree ranking tasks”—three

“that that (b) response”—extra “that”

We have corrected those errors accordingly.

Reviewer 2

R2-1 Of the 134 neuro email addresses, is there a possibility that some of these belong to a common person? That is, could some individual(s) have been represented multiple times in the responses, or were steps taken to somehow remove responses from an individual with multiple email addresses?

We checked for duplicate email addresses before sending out participation invitations. We also checked for duplicates responses with the typical means for any type of online survey process, i.e., evaluation of IP addresses, comparing the similarity of response patterns, etc. We have added short notes on that point on page 6.

R2-2 Of the 574 geographer email addresses, is there a possibility that some of these belong to a common person? That is, could some individuals have been represented multiple times in the responses, or were steps taken to somehow remove responses from an individual with multiple email addresses? The problem seems potentially more severe here (than with the neuro people) as there were also some addresses from ICA here.

There is indeed a very small, albeit existent possibility that target people may have received the survey invitation through multiple channels (as already mentioned in the paper). However, as mentioned above we did check for duplicates. We are confident that we did not analyze duplicate data sets from an identical single respondent. We have added this clarification on p. 6. We cannot fully exclude the very small, but existent, possibility of malicious responders (i.e., the same person responding to the survey from different computers and in different ways). We do consider this probability for such malicious behavior to be very low, as no renumeration of any sort for participation was offered to the experts.

R2-3 I wonder if some "master experts" who were contacted handed off their response to someone else. One way to test that would be to determine if any of the inputs came from email addresses different than those who were emailed (at least for the neuro people).

Yes, we acknowledge this possibility, not only for neuro people. However, this would not be problematic for us, as we did assess in detail the experience of each respondent. We thus might have received answers of a person that may have less (or even more) experience than the targeted individual we had contacted by Email. As we applied pair-matching of experts with respect to their experience in the statistical analysis, this would thus not have affected our result.

R2-4 How much were the Mechanical Turk people paid? How much were the domain experts paid?

We have added this information on page 6. We thank the reviewer for this question.

R2-5 The term "locked-in" in "Image Creation" is not well-known; a different term is needed.

Thank you for helping us clarify this point. We mentioned on page 7 that all terms were explained in the questionnaire, and we also have added a short explanation of the “locked in” state in the paper.

There is a typo in "Image Creation", Sentence 3: "graphic" -> "geographic".

Thank you – the typo has been corrected.

There are two grammar errors in "Image Creation": Sentence 6, "such as, residential" should be ", such as residential"; Sentence 7, "set is" should be "set was"

Thank you – the errors have been corrected.

R2-6 In "Structure of the online...", it should be mentioned if any of the Mechanical Turk participants were classified as experts? In addition, if any from the neuro or geography groups were classified as lay (or if none were), that should be mentioned.

We thank the reviewer for this observation. We added this information on page 6.

In "Structure of the online...", Paragraph 3, "likert" should be "Likert".

Thank you – the typo has been corrected.

R2-7 In "Structure of the online....", Paragraph 3, it should be made clear if the lay people answered all these questions.

We clarified on page 9 that indeed all persons answered those questions.

In "Structure of the online....", Paragraph 4, "more a consumer or a producer" should be "more consumers or producers".

Thank you – the error has been corrected.

In "Structure of the online....", Paragraph 4, what is "of main text" referring to? Is there another Figure 1 in another component that was not submitted?

Thank you – the error has been corrected, there is indeed no other “Figure 1”.

R2-8 In "Structure of the online....", Paragraph 6, Sentence 2, this point should also be made earlier--just before Paragraph 2 of "Structure of the online..." (or perhaps only made there).

We now have made this point already in the fourth paragraph of this section.

R2-9 Who were the 27 participants that tested questionnaire usability--were any of them the same ones that took part in the other parts of the study? Did any of the 27 that tested usability drop out due to the length of the survey completion?

On page 11 we clarified that all usability testers were domain experts and that no data of the pretest have been used in the main study.

R2-10 There is a sentence frag. just above Table 1 ("Specifically....").

Thank you – the error has been corrected.

R2-11 Figure 3 seems misleading as it aggregates the Table 1 information without regard to context. I'm not convinced of the conclusions related to such aggregated material...domain experts, especially if they have a long history in their domains, may tend to become very domain-centric and thus perhaps more likely to use their own training as they approach another domain, which may be a serious error for the other domain. I think the Figure 3 material should be de-aggregated, with the study then considering each context for each expert set, without considering "geo" as a whole, for example. Alternately, the focus could be just neuro experts on neuro data and geo experts just on geo data.

We thank the reviewer for this important remark, as it points to the fact that the explanation for Figure 3 was not clear enough. We added in the paper that Figure 3 does not aggregate any data from Table 1, but compares the general suitability ratings of color scales (assessed by each of the expert groups) 

R2-12 In "Factors potentially," the focus suddenly shifts to "preference." I don't see where that comes from. Moreover, the Figure 3 material is suitability versus trust, NOT preference versus trust. There is a difference between something being suitable and something being preferred. This part of the paper needs some re-thought and re-work.

We thank the reviewer for this important observation. We indeed did ask participants specifically about the “suitability” of one of the four presented scales to display the increase of a statistical parameter of any type (see survey question in the supplementary information). Being nonnative speakers we now realize that “suitability” cannot be equated with “preference”, thus we now correctly use the original term suitability systematically across the paper (in particular page 16). We report the original question as participants saw it in the questionnaire on page 10.

R2-13 The current Figure 3 is not convincing to me that "geovis experts seem to trust the heated body color scale most" (statement just before the "Other variables" section) - the trust levels are pretty close for all. Either a statistical showing there is needed or the statement needs to be removed.

Information on statistical trends has been added and the statement on page 17 has been adapted.

R2-14 Table 2 has a low information-to-space ratio. The table should be reformatted in some way to be made much more compact - nearly every entry there is "ns" but the table takes up about 5% of the entire page length of the paper.

Thank you for this comment. We agree, and make the table available in supplementary information.

p. 19, "Our hypotheses" paragraph, line 5, "that that" should be "that".

Thank you – the error has been corrected.

R2-15 Ref. 2 is incomplete.

We have replaced this reference by a very recently published paper (Crameri et al. 2020), which lead to a slight renumbering of the first 9 references.

Lastly, a final positive remark: Next-to-last paragraph last sentence makes a very good point!

We thank the reviewer for this positive remark.

---

## [Decision Letter · Decision Letter 1]

5 Jan 2021

PONE-D-20-23193R1

Susceptibility of domain experts to color manipulation indicate a need for design principles in data visualization

PLOS ONE

Dear Dr. Christen,

Your revised manuscript has now been reviewed by the two original reviewers. As you will see, they were generally happy with the revision, but reviewer #1 still has a few minor points that they would like to see addressed. Therefore, we are inviting you to submit another revision. Since the points are all very small, I will most likely not send the manuscript back to the reviewers after your revision, but instead render a final decision based on my own evaluation.

We look forward to receiving your revised manuscript.

Kind regards,

Ronald van den Berg

Academic Editor

PLOS ONE

Reviewers' comments:

Reviewer's Responses to Questions

**Comments to the Author**

1. If the authors have adequately addressed your comments raised in a previous round of review and you feel that this manuscript is now acceptable for publication, you may indicate that here to bypass the “Comments to the Author” section, enter your conflict of interest statement in the “Confidential to Editor” section, and submit your "Accept" recommendation.

Reviewer #1: All comments have been addressed

Reviewer #2: (No Response)

2. Is the manuscript technically sound, and do the data support the conclusions?

Reviewer #1: Yes

Reviewer #2: Yes

3. Has the statistical analysis been performed appropriately and rigorously? 

Reviewer #1: Yes

Reviewer #2: Yes

4. Have the authors made all data underlying the findings in their manuscript fully available?

Reviewer #1: Yes

Reviewer #2: Yes

5. Is the manuscript presented in an intelligible fashion and written in standard English?

Reviewer #1: Yes

Reviewer #2: Yes

6. Review Comments to the Author

Reviewer #1: I thank the authors for their thoughtful and thorough revision. The clarifications significantly improve the legibility of the paper and the connection between the conclusions and data. I have a few small points that may benefit from consideration and possible revision, but these are not points that would require rereview. The work will be an interesting addition to our understanding of color use in visualization.

To clarify the concerns about ceiling effects, my assumption based on the methods description of the prior paper had been that the scale ranged from 0-8 rather than 1-9. With a cap at 9, I agree that there are not likely ceiling effects. Thanks for the clarification.

The definition of trust makes the rest of the conclusions a lot more logical and easier to follow. Given the definition, the results seem better supported compared with more conventional definitions. I wonder whether “trust” is really what this metric captures (it seems more like a measure of the validity of the conclusions), but so long as a clear definition is maintained, this is a minor point that I simply want to raise for the authors to consider as a potential opportunity for further clarity. It would similarly be useful to reiterate this definition when introducing these measures in the Structure section.

As another minor point for clarification, simply being blue-white-red does not necessarily mean the scale will be color-blind friendly (it depends on the internal variations in each of the blue and red sections). It’s worth noting if the scale has been externally confirmed to be colorblind safe (e.g., by the source document or by an external simulator). Most such encodings are, but explicit validity would be useful here.

- “In the methods part”—Methods section

- “Nine, respectively ten persons”—It seems like there’s an issue here: is it nine or ten? Also, were these participants removed or reclassified as they don’t fit the definition of a “lay” participant?

- “color vision deficiencies (no deficiency… (i.e., color blindness)”—Missing closing parentheses

- Some of the paragraphs in the discussion are quite long. For legibility, it may be worth breaking these into smaller pieces.

Reviewer #2: This manuscript is a nice improvement on the initial submission. I recommend its acceptance. I did notice one small typo to fix - p. 22, line 3, "assessing" -> "assess"

I appreciate the nice work of the authors!

7. PLOS authors have the option to publish the peer review history of their article (what does this mean?). If published, this will include your full peer review and any attached files.

Reviewer #1: No

Reviewer #2: No

---

## [Author Response · Author response to Decision Letter 1]

12 Jan 2021

Susceptibility of domain experts to color manipulation indicate a need for design principles in data visualization – Response to 2. Review

We thank the reviewers for their detailed and constructive review. Below we respond to revision sug-gestions by reviewers in turn. All respective changes in the first revision iteration are marked in yellow, the revisions made in the second iteration are marked in green in the document “Revised Manuscript with Track Changes” (some smaller corrections in the yellow parts have not been marked in green).

Reviewer 1

The definition of trust makes the rest of the conclusions a lot more logical and easier to follow. Given the definition, the results seem better supported compared with more conventional definitions. I won-der whether “trust” is really what this metric captures (it seems more like a measure of the validity of the conclusions), but so long as a clear definition is maintained, this is a minor point that I simply want to raise for the authors to consider as a potential opportunity for further clarity. It would similarly be useful to reiterate this definition when introducing these measures in the Structure section.

We have further clarified our definition of “trust” on page 4 and reiterated this clarification on page 9.

As another minor point for clarification, simply being blue-white-red does not necessarily mean the scale will be color-blind friendly (it depends on the internal variations in each of the blue and red sec-tions). It’s worth noting if the scale has been externally confirmed to be colorblind safe (e.g., by the source document or by an external simulator). Most such encodings are, but explicit validity would be useful here.

On page 8, we added detailed information that the bivariate blue-white-red scale has been checked for colorblind safety.

Small issues:

- “In the methods part”—Methods section

- “Nine, respectively ten persons”—It seems like there’s an issue here: is it nine or ten? Also, were these participants removed or reclassified as they don’t fit the definition of a “lay” participant?

- “color vision deficiencies (no deficiency… (i.e., color blindness)”—Missing closing parentheses

- Some of the paragraphs in the discussion are quite long. For legibility, it may be worth breaking these into smaller pieces.

We thank the reviewer for this observation. We have addressed all issues accordingly.

Reviewer 2

This manuscript is a nice improvement on the initial submission. I recommend its acceptance. I did no-tice one small typo to fix - p. 22, line 3, "assessing" -> "assess"

We thank the reviewer for this observation. We have corrected the typo accordingly.

---

## [Editor Report · Decision Letter 2]

20 Jan 2021

Susceptibility of domain experts to color manipulation indicate a need for design principles in data visualization

PONE-D-20-23193R2

Dear Dr. Christen,

We’re pleased to inform you that your manuscript has been judged scientifically suitable for publication and will be formally accepted for publication once it meets all outstanding technical requirements.

Kind regards,

Ronald van den Berg

Academic Editor

PLOS ONE
---

## [Editor Report · Acceptance letter]

25 Jan 2021

PONE-D-20-23193R2 

Susceptibility of domain experts to color manipulation indicate a need for design principles in data visualization 

Dear Dr. Christen:

I'm pleased to inform you that your manuscript has been deemed suitable for publication in PLOS ONE. Congratulations! Your manuscript is now with our production department. 

Kind regards, 

on behalf of

Dr. Ronald van den Berg 

Academic Editor

PLOS ONE